# Early detection of cardiotoxicity in children receiving adriamycin: A comparison of tissue Doppler and standard echocardiography with Troponin I as an early marker

Mina Farshidgohar[1], Majid Vafaie[1], Sonia Oveisi[2]*, Maryam Bakhshi[2]

**1** Clinical Research Development Unit of Advanced Medicine, Qazvin University of Medical Science, Qazvin, Iran, **2** Non-communicable Diseases Research Center, Research Institute for Prevention of Non-communicable Diseases, Qazvin University of Medical Sciences, Qazvin, Iran

* soniaoveisi@gmail.com

## Abstract

### Background

Adriamycin (doxorubicin hydrochloride) is a widely used chemotherapeutic agent for treating various malignancies. However, its most significant adverse effect is cardiac toxicity. Early detection of Adriamycin-induced cardiac dysfunction is crucial in preventing heart failure through interventions such as angiotensin-converting enzyme inhibitors and beta-blockers. Adriamycin-induced cardiotoxicity is classified into Type I (irreversible, involving cardiomyocyte necrosis) and Type II (potentially reversible, involving cardiomyocyte dysfunction). Type I toxicity is more common, leading to long-term heart cell necrosis. Monitoring cardiac function in children receiving Adriamycin is essential. Non-invasive imaging techniques like transthoracic echocardiography, cardiac magnetic resonance imaging, and computed tomography are used to evaluate left ventricular (LV) systolic and diastolic functions. However, the role of cardiac Troponin I for early detection of anthracycline-induced cardiomyopathy remains debated.

### Methods

This prospective study included 50 children (26 males, 24 females) with a median age of 8 years (range: 2–14) receiving Adriamycin. Troponin I levels were measured before and 24 hours after Adriamycin administration. Standard Doppler Echocardiography and Tissue Doppler Imaging (TDI) were performed at baseline, one month, and six months' post-treatment to assess LV function.

### Results

Out of 50 patients, two showed a decrease in left ventricular ejection fraction (LVEF) below the normal range (<55%) one month post-treatment, along with significant

**Data availability statement:** Data cannot be shared publicly due to confidentiality restrictions involving patient privacy. However, data are available from the Ethics Committee of Qazvin University of Medical Sciences for researchers who meet the institutional criteria for access to confidential data. Requests for access to the data may be directed to Afshin Mehralian, Administrative Secretary of the Ethics Committee, at the Ethics Committee of Qazvin University of Medical Sciences, via ethics@qums.ac.ir, referencing the project code IR.QUMS.REC.1400.090.

**Funding:** The author(s) received no specific funding for this work.

**Competing interests:** The authors have declared that no competing interests exist.

**Abbreviations:** LV, Left Ventricle; LVEF, Left Ventricular Ejection Fraction; TDI, Tissue Doppler Imaging; RV, Right Ventricle; TAPSE, Tricuspid Annular Plane Systolic Excursion; S' or SM, Systolic velocity (measured by Tissue Doppler Imaging); E', Early diastolic velocity (measured by Tissue Doppler Imaging); E/E', The ratio of early diastolic mitral inflow velocity (E) to early diastolic mitral annulus velocity (E') (measured by Tissue Doppler Imaging); E/A, The ratio of early (E) to late (A) ventricular filling velocities

increases in Troponin I levels. No significant decline in LVEF was observed at one or six months' post-treatment using Standard Doppler Echocardiography. However, TDI revealed a decrease in LV systolic and diastolic function in all patients one month after Adriamycin administration.

## Conclusion

TDI is more sensitive than Standard Doppler Echocardiography in detecting Adriamycin-induced cardiotoxicity. Elevated cardiac Troponin I levels correlate with a decline in LVEF, but subclinical cardiac dysfunction is better detected with TDI.

## Introduction

Adriamycin (doxorubicin hydrochloride) is a potent chemotherapeutic agent used to treat a wide range of cancers, including leukemia, lymphoma, and solid tumors. However, its most significant limitation is its potential to cause severe cardiac toxicity [1]. The most serious consequence of Adriamycin-induced toxicity is heart failure, which can be prevented or mitigated if detected early. Early identification of Adriamycin-induced cardiac dysfunction is crucial for initiating timely interventions, such as the use of angiotensin-converting enzyme inhibitors, beta-blockers, or other cardioprotective therapies.

Adriamycin-induced cardiotoxicity is classified into two types: Type I and Type II [2].Type I toxicity is irreversible and involves cardiomyocyte necrosis, while Type II toxicity is potentially reversible, involving transient cardiomyocyte dysfunction. Adriamycin predominantly induces Type I cardiotoxicity, leading to long-term cardiac damage and cell necrosis [2].

Monitoring cardiac function in children receiving Adriamycin is essential for detecting and managing potential cardiotoxic effects. Traditional non-invasive imaging techniques, such as transthoracic echocardiography, cardiac magnetic resonance imaging, and computed tomography, are commonly used to evaluate left ventricular (LV) systolic and diastolic function [3,4].

While these techniques provide valuable information, the role of cardiac Troponin I as an early biomarker for anthracycline-induced cardiomyopathy remains controversial [5].

This study aims to assess the effectiveness of Troponin I as an early marker of Adriamycin-induced cardiotoxicity and compare the efficacy of Standard Doppler Echocardiography and Tissue Doppler Imaging (TDI) in evaluating cardiac function in pediatric patients undergoing Adriamycin treatment.

## Methods and materials

### Study design

This prospective study was conducted at a tertiary pediatric oncology center in Iran from February to September 2024. The study included 50 pediatric patients diagnosed with cancer who were undergoing Adriamycin treatment. Inclusion criteria

were all pediatric cancer patients (with leukemia, lymphoma, or solid tumors). Exclusion criteria included congenital heart disease, prior mediastinal irradiation, and withdrawal from the study.

The current investigation was conducted in accordance with the principles and recommendations outlined in the Helsinki Declaration. The utmost confidentiality was maintained for all collected data. The ethics committee of Qazvin University of Medical Sciences approved the project; assigning it the ethics code IR.QUMS.REC.1400.090. Since the participants in this study included minors, written informed consent was obtained from all parents.

### Measures

Echocardiographic assessments were performed at baseline, one month, and six months after the administration of Adriamycin. Serum Troponin I levels were measured before and 24 hours' post-treatment. Both systolic and diastolic left ventricular (LV) functions were evaluated using Tissue Doppler Imaging (TDI) and Standard Doppler Echocardiography [6].

### Echocardiographic parameters

**Left Ventricular Ejection Fraction (LVEF):** Normal LVEF was defined as ≥55%, and abnormal LVEF as <55% [7]. **Diastolic Function:** Normal diaslic function was defined as an E/A ratio of 0.9–1.5. Abnormal diastolic function was considered if the E/A ratio was < 0.9 or >2 [8].

### Tissue doppler parameters

- **S' (Systolic Velocity):** Systolic myocardial velocity at the lateral mitral annulus.

- **E' and A' (Diastolic Velocities):** Early and late diastolic velocities.

- **E/E' Ratio:** An indicator of left ventricular filling pressure [9].

TAPSE (Tricuspid Annular Plane Systolic Excursion): Normal TAPSE was defined as >1.9 cm; reduced systolic function was indicated by TAPSE <1.5 cm [10].

In the TDI method, the lateral segment of the mitral annulus was identified as the optimal location for the sample volume (Fig 1) [11].

### Statistical analysis

Descriptive statistics (mean ± standard deviation) were used to summarize quantitative variables, while categorical variables were reported as frequency (n) and percentage (%). To compare systolic and diastolic parameters (S', E', E/A, E/E', TAPSE), the analysis of variance (ANOVA) was applied. The chi-square test ($\chi^2$) was used for categorical variables such as LVEF and TAPSE. Categorical variables were defined as follows:

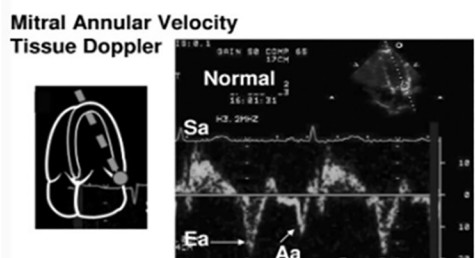

**Fig 1. Parameters of Tissue Doppler Echocardiography.**

- **LVEF:** ≥ 55% = 1, 45–54% = 2, 35–44% = 3.

- **TAPSE:** > 2 cm = 1, 1.5–2 cm = 2, < 1.5 cm = 3.

- **S':** ≥ 8 cm/s = 1, < 8 cm/s = 2.

- **E/E':** > 8 = 2, ≤ 8 = 1.

- **E':** ≥ 18 cm/s = 1, < 18 cm/s = 2.

A p-value of <0.05 was considered statistically significant. Data analysis was performed using SPSS software version 25.

## Results

### Demographics

This study included a total of 50 pediatric patients, comprising 26 males (52%) and 24 females (48%), with a median age of 8 years (range: 2–14 years). Patients were diagnosed with various malignancies, including leukemia (n = 30, 60%), lymphoma (n = 9, 18%), and solid tumors (n = 11, 22%). The median cumulative dose of Adriamycin administered was 400 mg/m², with individual doses adjusted based on treatment protocols and patient-specific factors. The duration of chemotherapy regimens varied depending on the type and stage of malignancy, with treatment spanning several months. Patients were monitored for cardiotoxic effects using serial echocardiographic assessments and biomarker measurements at baseline, one month, and six months' post-treatment. The median follow-up duration for cardiac evaluation was six months, ensuring a comprehensive assessment of both early and subclinical Adriamycin-induced cardiotoxicities.

### Electrocardiographic findings

At baseline, all electrocardiograms (ECGs) were within normal limits. However, at one-month follow-up, 20% of the patients (n = 10) showed signs of sinus tachycardia, while 12% (n = 6) had decreased QRS voltage, but no serious arrhythmias were noted. By the six-month follow-up, sinus tachycardia persisted in 10% of the patients (n = 5), and decreased QRS voltage remained in 8% (n = 4). These ECG findings suggest that Adriamycin-induced cardiotoxicity can lead to subtle, early changes in the heart's electrical activity, potentially preceding more obvious functional impairments.

### Troponin I levels

Significant increases in Troponin I levels were observed in two patients, which correlated with a decrease in left ventricular ejection fraction (LVEF) to below the normal range (<55%). In response, cardioprotective therapy was initiated with captopril and a beta-blocker, while chemotherapy, including Adriamycin, was continued without discontinuation. These patients remained under close echocardiographic surveillance to monitor cardiac function during treatment. During follow-up, both patients remained on captopril and beta-blocker therapy while continuing chemotherapy. Notably, their cardiac function remained stable, with no further deterioration observed.

In the remaining 48 patients, despite normal Troponin I levels, Tissue Doppler Imaging (TDI) revealed subclinical left ventricular (LV) systolic and diastolic dysfunction.

### Standard Doppler echocardiography

Due to the data not adhering to a normal distribution (Kolmogorov–Smirnov test, p < 0.05), LVEF values were presented as medians and interquartile ranges. Table 1 illustrates LVEF slightly declined over time, and the Friedman test confirmed a significant change at one and six months after Adriamycin treatment (p < 0.001).

Tissue Doppler Imaging (TDI) findings: Significant reductions in systolic (S') and diastolic (E') velocities were observed (p < 0.001). The mean S' velocity before starting Adriamycin treatment, one month after treatment, and six months after

**Table 1. Data of Echocardiography parameters of the Tissue Doppler in three stages: Before, 1 month and 6 months after taking Adriamycin.**

|  | Baseline | 1-Month | 6-Months | f | p-value |
|---|---|---|---|---|---|
| EF | 65 (50, 78) * | 65 (50, 75) * | 62 (50, 75) * | 14.42 | P=0.001 |
| SM | 10.29±1.41 (9.9, 10,7) | 8.30±1.31 (7.9, 8.7) | 8.35±1.23 (8.0, 8.7) | 37.01 | P<0.001 |
| E' | 16.87±2.28 (16.2, 17.52) | 13.98±2.90 (13.15, 14.8) | 14.85 (3.53) * | 19.16 | P<0.001 |
| E/A | 1.77±0.38 (1.67, 1.88) | 1.53±0.43 (1.41, 1.65) | 1.5 (0.35) * | 6.17 | P=0.033 |
| TAPSE | 2.19±0.5 (2.05, 2.33) | 1.79±0.46 (1.66, 1.92) | 1.80±0.46 (1.67, 1.93) | 11.72 | P<0.001 |
| E/E' | 7.2 (1.58) * | 8.7 (1.57) * | 8.5 (1.65) * | 8.95 | P<0.001 |

* Data that do not follow a normal distribution was presented as medians and interquartile ranges (Q1-Q3).

treatment was 10.3 cm/s, 8.3 cm/s, and 8.3 cm/s, respectively (p<0.001). The mean E' velocity before starting Adriamycin treatment, one month and six months after treatment was 16.8 cm/s,14 cm/s and 14 cm/s (p<0.001).

The E/E' ratio increased significantly from 7.6 to 9.4 (p<0.001), indicating worsening diastolic function. Right ventricular (RV) TAPSE showed a significant decline (p<0.001), further supporting evidence of myocardial dysfunction. The mean RV TAPSE before starting Adriamycin treatment, one month after treatment, and six months after treatment was 2.2 cm, 1.7 cm, 1.8 cm (P<0.001) (Table 1). Additionally, the frequency of categorical variables was analyzed and compared across the three stages using the Chi-square test (X²) (Table 2). Abnormal E/E' ratios (≥10) were observed in 80% of patient's one-month post-treatment, compared to only 24% before treatment (Table 2), indicating a significant decline in left ventricular diastolic function. Tissue Doppler Echocardiography also revealed a marked decrease in left ventricular systolic function in all patients. Subclinical left ventricular systolic dysfunction (S'<8 cm/s) was detected in 32% of patients one month after treatment, whereas no patients had an S' velocity below 8 cm/s prior to Adriamycin administration (Table 2).

**Table 2. Frequency (%) of categorical variables to compare in three stages by X².**

|  | Baseline N (%) | 1-Month N (%) | 6-Months N (%) | X² | p-value |
|---|---|---|---|---|---|
| E' |  |  |  | 20.37 | P<0.001 |
| < 8 cm/s | 35 (70) | 48 (96) | 48 (96) |  |  |
| ≥ 8 cm/s | 15 (30) | 2 (4) | 2 (4) |  |  |
| E/E' |  |  |  | 46.01 | P<0.001 |
| < 8 cm/s | 38 (76) | 10 (20) | 9 (18) |  |  |
| ≥ 8 cm/s | 12 (24) | 40 (80) | 41 (82) |  |  |
| SM |  |  |  | 19.6 | P<0.001 |
| < 8 cm/s | 0 (0) | 16 (32) | 15 (30) |  |  |
| ≥ 8 cm/s | 50 (100) | 34 (68) | 35 (70) |  |  |
| Ejection Fraction |  |  |  | 3.8 | P=0.149 |
| 35 - 44 | 0 (0) | 0 (0) | 0 (0) |  |  |
| 45 - 54 | 3 (6) | 2 (4) | 7 (14) |  |  |
| ≥ 55 cm/s | 47 (94) | 48 (96) | 43 (86) |  |  |
| RV TAPSE |  |  |  | 18.02 | P=0.01 |
| < 1.5 cm/s | 0 (0) | 12 (24) | 11 (22) |  |  |
| 1.5 - 2 | 17 (34) | 19 (38) | 21 (42) |  |  |
| ≥ 2 cm/s | 33 (66) | 19 (38) | 18 (36) |  |  |
| Total | 50 (100) | 50 (100) | 50 (100) |  |  |

Similarly, no patients had an RV TAPSE below 15 mm before treatment; however, one month after Adriamycin initiation, 24% of patients exhibited an RV TAPSE below 1.5 cm (Table 2). These findings indicate deterioration in right ventricular function within the first month following Adriamycin treatment.

## Discussion

The emergence of heart failure in cancer survivors, particularly after treatment with anthracyclines like Adriamycin, remains a major cause of morbidity and mortality [12]. Belham and his colleagues evaluated left ventricular (LV) function in adults receiving anthracycline therapy. They found that 26% of patients, despite lacking significant pre-existing cardiac disease, developed cardiotoxicity [13]. Cardiovascular complications, including early cardiac dysfunction, are often subtle and may not be apparent through traditional assessments, which makes early detection crucial for effective management. In our study, we observed a clear pattern that supports the necessity for vigilant cardiac monitoring in children undergoing Adriamycin therapy.

Bayram et al. investigated cardiotoxicity in childhood leukemia survivors who had received low-dose anthracycline therapy, with a particular focus on systolic and diastolic myocardial function assessed through conventional echocardiography and Tissue Doppler Imaging (TDI). Their findings demonstrated the superiority of TDI in detecting early cardiotoxicity before significant reductions in LVEF, which is consistent with our study. They reported diastolic dysfunction in their patient cohort and identified significant alterations in myocardial velocities ($S^m$, $E^m$, and $A^m$) using TDI. In alignment with these findings, our study observed early diastolic dysfunction in 80% of patients at one-month post-treatment [14], further emphasizing the clinical utility of TDI in the early detection of anthracycline-induced cardiotoxicity.

Our findings suggest that while Troponin I is an important biomarker for detecting myocardial injury, it may not be sensitive enough to detect early subclinical cardiac dysfunction in all patients. This observation aligns with recent studies suggesting that Troponin I elevations are not always reflective of early systolic or diastolic dysfunction [15,16]. Notably, in our cohort, while two patients with significantly elevated Troponin I levels showed a decrease in LVEF, the majority of patients, even with normal Troponin I levels, exhibited subtle but significant reductions in systolic and diastolic function when assessed by Tissue Doppler Imaging (TDI). This disparity highlights the limitations of relying solely on Troponin I levels for early detection of Adriamycin-induced cardiotoxicity.

Interestingly, despite the absence of Troponin I elevation in most patients, TDI was able to identify subclinical changes in both systolic and diastolic functions. This underscores the superior sensitivity of TDI compared to standard Doppler echocardiography in detecting early cardiac dysfunction. These findings are in line with recent discussions on the need to incorporate advanced imaging techniques, like TDI, in routine surveillance of pediatric cancer patients who receive cardiotoxic treatments. Standard Doppler echocardiography, by focusing primarily on LVEF, might overlook these early, subtle changes that are essential for timely intervention.

A potential explanation for the lack of Troponin I elevation in most patients could be related to the timing of measurement. As Troponin I is a marker of myocardial injury, it is likely that its levels may rise more gradually, or at later time points, especially in cases of subclinical dysfunction. Our study measured Troponin I levels 24 hours' post-treatment, but future studies should consider longer intervals of observation to capture potential late-phase elevations in Troponin I and other biomarkers, which might correlate better with long-term cardiac outcomes.

A comparison with the study conducted by Venturelli et al. in 2018 further supports our findings. Both studies demonstrate the presence of diastolic dysfunction with preserved LVEF in patients treated with anthracyclines. However, their retrospective design with a median follow-up of two years differs from our prospective approach, which included serial assessments at baseline, one month, and six months' post-treatment. The longitudinal nature of our study provides a more detailed analysis of cardiac function changes over time. A key distinction between our study and that of Venturelli et al. is the incorporation of cardiac Troponin I measurements as a biochemical marker for cardiotoxicity. Our results indicate that elevated Troponin I levels correlate with a decline in LVEF [17].

Another important finding in our study was the detection of subclinical LV systolic dysfunction in 32% of patients by TDI, despite normal LVEF values. This suggests that relying on LVEF alone to define cardiotoxicity might miss earlier forms of myocardial impairment. It is important to note that even mild reductions in myocardial function detected early on can be significant, as they might predispose the heart to further damage over time, especially with continued exposure to cardiotoxic agents.

Furthermore, the observation of significant changes in the E/E' ratio, with 80% of patients showing abnormalities one month after receiving Adriamycin, compared to only 24% before treatment, adds further weight to the argument for using TDI to assess both systolic and diastolic functions. Diastolic dysfunction, often a precursor to more severe cardiac pathology, was identified at an early stage in the majority of our patients, suggesting that TDI provides a comprehensive approach to monitoring cardiac health in this vulnerable population.

In Study by Francesco Venturelli et al found diastolic impairment in childhood cancer survivors treated with anthracyclines, but did not focus specifically on the E/E' ratio. It did report abnormalities in diastolic function measurements like IVRT, DT, E, and A, but did not mention the percentage of patients affected or early detection of diastolic dysfunction. Also another studies did not specifically mention changes in the E/E' ratio or provide statistical evidence for early diastolic dysfunction in the early phases after anthracycline treatment in children.

While Adriamycin's ability to cause long-term reductions in LVEF is well documented, our study suggests that the early stages of cardiotoxicity might manifest as changes in myocardial velocities and diastolic function rather than overt changes in LVEF. These findings emphasize the importance of regular and early cardiac assessments using advanced echocardiographic techniques to identify subtle dysfunction before irreversible damage occurs. Given that many of these changes are reversible with early intervention, such as the use of beta-blockers or ACE inhibitors, early detection could significantly improve patient outcomes.

In our cohort, TDI was able to identify early subclinical changes that were not detectable by standard Doppler or Troponin I levels, highlighting the superior sensitivity of TDI in detecting early myocardial injury. By contrasting Troponin I levels with TDI findings, we reveal that TDI can detect functional abnormalities even in the absence of Troponin I elevation, adding an important layer of diagnostic capability that enhances early intervention strategies for preventing long-term cardiac complications.

Our study highlights the value of Tissue Doppler Echocardiography in the early detection of Adriamycin-induced cardiotoxicity, showing that it is more sensitive than standard Doppler echocardiography in identifying both systolic and diastolic dysfunction in pediatric patients. We advocate for the incorporation of TDI into routine clinical practice for monitoring these patients, as early intervention may prevent the progression of heart failure and improve long-term cardiac health. Further studies with longer follow-up periods are needed to better understand the long-term effects of Adriamycin on cardiac function and the role of advanced imaging techniques in detecting subclinical dysfunction.

Novelty our study compared to other studies is the inclusion of Troponin I as a biomarker for cardiotoxicity. In addition, our study focused on a pediatric population, whereas the most studies involved adults, making our findings relevant to a different patient demographic with potentially different responses to Adriamycin. E/E' Ratio as a Sensitive Marker: The E/E' ratio is a reliable marker of diastolic function, and the fact that 80% of patients showed abnormalities within a short time frame post-treatment makes it a valuable parameter for detecting early cardiac dysfunction. This early identification of diastolic dysfunction sets our study apart from the others, as the other studies either did not assess this or did not find such early abnormalities. Present study assessed cardiac function at one month and six months' post-treatment, providing insights into the early onset of cardiotoxicity. In summary, the novelty of our work lies in its comprehensive approach, combining Troponin I and TDI to assess both biomarkers and functional changes in a pediatric cohort treated with Adriamycin. The added value of our study lies in its early detection of subclinical cardiotoxicity, emphasizing the need for sensitive, non-invasive monitoring techniques like TDI to prevent long-term cardiac complications in pediatric cancer survivors.

## Limitations

This study has several limitations that should be acknowledged. First, while echocardiography, particularly Tissue Doppler Imaging (TDI), provided valuable insights into subclinical cardiac dysfunction, it has inherent limitations, including operator dependency and potential measurement variability. The absence of cardiac magnetic resonance imaging (CMR), which is considered the gold standard for myocardial assessment, limits the comprehensive evaluation of myocardial injury. Second, Troponin I levels were measured only immediately after the start of treatment, without serial assessments over time, which may have restricted the ability to detect late-onset cardiotoxicity. Additionally, the study lacked long-term follow-up data, preventing an evaluation of whether early echocardiographic or biochemical changes translated into persistent or progressive cardiac dysfunction. Future studies incorporating longer follow-up durations, serial biomarker assessments, and advanced imaging modalities are needed to further elucidate the long-term impact of Adriamycin-induced cardiotoxicity in pediatric patients.

## Conclusion

Our study highlights the superior sensitivity of Tissue Doppler Imaging (TDI) over standard Doppler echocardiography in detecting Adriamycin-induced cardiotoxicity. Elevated cardiac Troponin I levels correlate with a decline in LVEF, but subclinical cardiac dysfunction is better detected with TDI. Given the promising results of TDI in early detection, future studies could explore the potential impact of heart failure medications, such as ACE inhibitors (ACEi) or beta-blockers, on improving TDI parameters and overall cardiac function in patients receiving Adriamycin. Assessing how these therapeudtic agents may prevent or reverse TDI-detected dysfunction could provide valuable insights for optimizing treatment protocols and improving long-term cardiac health in pediatric cancer patients.

## Acknowledgments

The authors would like to thank the Clinical Research Development Unit, Qods Hospital, Qazvin University of Medical Sciences, Qazvin, Iran.

## Author contributions

**Conceptualization:** Mina Farshidgohar.

**Data curation:** Mina Farshidgohar, Maryam Bakhshi.

**Formal analysis:** Sonia Oveisi.

**Methodology:** Majid Vafaie, Sonia Oveisi.

**Project administration:** Sonia Oveisi.

**Software:** Sonia Oveisi.

**Supervision:** Mina Farshidgohar, Majid Vafaie.

**Validation:** Majid Vafaie.

**Writing – original draft:** Mina Farshidgohar, Maryam Bakhshi.

**Writing – review & editing:** Mina Farshidgohar, Sonia Oveisi, Maryam Bakhshi.

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
