## [Decision Letter · Decision Letter 0]

18 Mar 2025

Dear Dr. Oveisi,

Thank you for submitting your manuscript to PLOS ONE. After careful consideration, we feel that it has merit but does not fully meet PLOS ONE’s publication criteria as it currently stands. Therefore, we invite you to submit a revised version of the manuscript that addresses the points raised during the review process.

Thank you for your patience, it was difficult to find reviewers for your article, but one reviewer did a fairly thorough and detailed review. I am looking forward to a revised version of your manuscript.  

We look forward to receiving your revised manuscript.

Kind regards,

Dr. Benjamin Benzon, Ph.D., M.D.

Academic Editor

PLOS ONE

Journal Requirements:

2. In the ethics statement in the Methods, you have specified that verbal consent was obtained. Please provide additional details regarding how this consent was documented and witnessed, and state whether this was approved by the IRB

5. Please note that your Data Availability Statement is currently missing the DOI/accession number of each dataset OR a direct link to access each database. If your manuscript is accepted for publication, you will be asked to provide these details on a very short timeline. We therefore suggest that you provide this information now, though we will not hold up the peer review process if you are unable.

Additional Editor Comments :

Thank you for your patience, it was very hard to find a reviewers for your paper, however a one reviewer made a pretty thorough and specific review. I am looking forward to receiving a revised version of manuscript.

Reviewers' comments:

Reviewer's Responses to Questions

**Comments to the Author**

1. Is the manuscript technically sound, and do the data support the conclusions?

Reviewer #1: Yes

2. Has the statistical analysis been performed appropriately and rigorously?

Reviewer #1: Yes

3. Have the authors made all data underlying the findings in their manuscript fully available?

Reviewer #1: No

4. Is the manuscript presented in an intelligible fashion and written in standard English?

Reviewer #1: Yes

***Reviewer #1:*** I have read with great interest the paper "Early Detection of Cardiotoxicity in Children Receiving Adriamycin: A Comparison of Tissue Doppler and Standard Echocardiography with Troponin I as an Early Marker" by Farshidgohar et al. The manuscript explores the utility of echocardiographic methods, specifically tissue Doppler imaging (TDI), for the early identification of cardiac toxicity. Below are specific comments intended to enhance the clarity and depth of the manuscript.

In general, the manuscript would greatly benefit from a more comprehensive review of existing literature regarding the use of TDI and other echocardiographic measures to detect cardiotoxicity during oncology treatment (see suggested references below). Additionally, the authors should provide more detailed descriptions regarding the clinical course of the patients, specifically highlighting how abnormal TDI findings impacted clinical management. The manuscript would also be strengthened by including figures illustrating the echocardiographic changes due to treatment.

Methods

• The sentences: "The study included 50 pediatric patients diagnosed with cancer (leukemia, lymphoma, and solid tumors) who were undergoing Adriamycin treatment," and "The median cumulative dose of Adriamycin administered was 400 mg/m²," belong in the Results section. The Methods section should solely describe the study procedures.

• Please expand upon the inclusion criteria: Were all pediatric cancer patients (with leukemia, lymphoma, or solid tumors) approached, or were there additional selection criteria?

Results

• The "Echocardiographic Findings" section should be converted into paragraph form with complete sentences to align better with the style of an academic paper.

• For the statements: "Significant reductions in S’ and E’ velocities (p<0.001)," and "Right ventricular (RV) TAPSE showed a significant decline," please include specific numerical values.

• The sentence: "No significant changes in LVEF were observed one and six months post-treatment (mean LVEF: 64%, 63%, 62%, respectively)" requires the addition of 95% confidence intervals (CIs) for these means if the data distribution was indeed normal. If not normally distributed, please convert to median and IQR.

• In the sentence "In Tissue Doppler echocardiography, a significant decrease in the systolic function of the left ventricle was observed in all patients.” Please be more specific and provide actual results and Chi-square test results.

• The sentence “Additionally, we analyzed the frequency (%) of categorical variables to compare them across three stages using the Chi-square test (X²) (Table 2) belongs in the Methods section, whereas here you should present the findings derived from these analyses in the results.

Additional clarifications requested:

• Describe the clinical management of the patients included in the study, with focus on patients with abnormal echo / troponin values. Was Adriamycin discontinued for any patient? Did you initiate cardioprotective therapy (ACE inhibitors or beta-blockers)? Provide more clinical details, particularly for the two patients with elevated Troponin I levels: Did they continue treatment?

• Expand the first paragraph of the Results to include cohort demographics, detailed oncologic diagnoses, specific treatment durations, Adriamycin dosage, and follow-up durations.

• Were additional follow-up diagnostic tests (such as cardiac MRI or stress testing) conducted for any patients, especially those with abnormal echo results?

• Report any abnormal ECG findings (e.g., sinus tachycardia, decreased voltage, QT prolongation).

• Include a figure and/or video (can be in the supplementary material) demonstrating echocardiographic changes before and after treatment.

• Provide definitions for all abbreviations used in the manuscript (e.g., "SM").

Discussion

• Introduce a dedicated "Limitations" paragraph, addressing issues such as the lack of additional diagnostic cardiac tests, limitations and potential inaccuracies associated with the echocardiographic methods used, and absence of long-term follow-up data. In this paragraph you can also include the limitation of having troponin only immediately after the start of treatment.

• Incorporate references to other relevant studies, discussing how your findings relate to previous research and highlighting the novelty and added value of your work compared to existing evidence. Suggested references include:

https://pmc.ncbi.nlm.nih.gov/articles/PMC7048119/

https://pubmed.ncbi.nlm.nih.gov/25577226/

https://pubmed.ncbi.nlm.nih.gov/22825896/

https://journals.lww.com/anti-cancerdrugs/abstract/2011/06000/chemotherapy_induced_cardiotoxicity__role_of_the.12.aspx

https://onlinelibrary.wiley.com/doi/10.5402/2012/870549

https://bjcardio.co.uk/2016/04/strain-imaging-and-anthracycline-cardiotoxicity/

**Do you want your identity to be public for this peer review?** For information about this choice, including consent withdrawal, please see our Privacy Policy

Reviewer #1: No

---

## [Author Response · Author response to Decision Letter 1]

15 Apr 2025

1. Your ethics statement should only appear in the Methods section of the manuscript. If Ethics statement is written in any section besides the Methods, please move it to the Methods section and delete it from any other section. Please ensure that your ethics statement is included in your manuscript, as the ethics statement entered into the online submission form will not be published alongside your manuscript.

Responce: Our ethics statement only was appeared in the Methods section of the manuscript.

Responce: We presented additional details regarding participant consent.

---

## [Decision Letter · Decision Letter 1]

27 Apr 2025

Dear Dr. Oveisi,

Thank you for submitting your manuscript to PLOS ONE. After careful consideration, we feel that it has merit but does not fully meet PLOS ONE’s publication criteria as it currently stands. Therefore, we invite you to submit a revised version of the manuscript that addresses the points raised during the review process.

You have made a good progress in answering the first round of reviews and the manuscript is much better now, please revise your manuscript according to the comments made by the reviewer in a second round.

We look forward to receiving your revised manuscript.

Kind regards,

Benjamin Benzon, Ph.D., M.D.

Academic Editor

PLOS ONE

Journal Requirements:

**Additional Editor Comments:**

You have made a good progress in answering the first round of reviews and the manuscript is much better now, please revise your manuscript according to the comments made by the reviewer in a second round.

Reviewers' comments:

Reviewer's Responses to Questions

**Comments to the Author**

Reviewer #1: (No Response)

2. Is the manuscript technically sound, and do the data support the conclusions?

Reviewer #1: Partly

3. Has the statistical analysis been performed appropriately and rigorously?

Reviewer #1: Yes

4. Have the authors made all data underlying the findings in their manuscript fully available?

Reviewer #1: No

5. Is the manuscript presented in an intelligible fashion and written in standard English?

Reviewer #1: Yes

Reviewer #1: Review (RR1) – Early Detection of Cardiotoxicity in Children Receiving Adriamycin: A Comparison of Tissue Doppler and Standard Echocardiography with Troponin I as an Early Marker

I thank the authors for addressing the previous comments and making notable improvements to the manuscript. However, there remain some important areas needing further clarification and revision to enhance the manuscript's overall quality. Specific points are detailed below.

Introduction

No comments

Methods

• “Diastolic Function: Normal diastolic function was defined as an E/A ratio of 0.9–1.5. Abnormal diastolic function was considered if the E/A ratio was <0.9 or >2.” Clarify how values between 1.5 and 2 were categorized, as the provided ranges do not currently cover these values. Did you intend the normal range to be 0.9–2?

• All echocardiographic variables you reference (LVEF, TAPSE, S’, E/E’, E/A) vary significantly by age across the 2–14 year range in your study. Applying adult-reference ranges universally is a substantial limitation and should be acknowledged explicitly in the discussion. You may, however, also mention that since comparisons were intra-patient (patients compared to themselves), then this limitation likely does not significantly impact your main results.

Results

• “The median follow-up duration for cardiac evaluation was six months.” Given that all patients had a minimum follow-up of 6 months, stating the median as exactly 6 months is unclear or possibly incorrect. Consider revising to: "All patients were followed for the 6-month duration of the study, and no further follow-up data were available," or clearly state the actual median and IQR for available follow-up durations.

• “Standard Doppler Echocardiography:” Please correct the formatting to properly align the section title. (remove the bullet-point format and align the paragraph with the rest of the text).

• The presentation of the statistical analysis results is currently difficult to follow. Clarify precisely what EF1, EF2, and EF3 represent. Consider revising to: "The Kolmogorov–Smirnov test indicated significant deviation from a normal distribution (statistic > 0.153, p < 0.05). The median EF at baseline, one month, and six months post-treatment were 65% (range: 50–78%), 65% (range: 50–75%), and 62% (range: 50–76%), respectively."

• “The mean S’ velocity before starting Adriamycin treatment, one month after treatment, and six months after treatment was 10.3 cm/s, 8.3 cm/s, and 8.3 cm/s,” and similarly, “The mean E’ velocity before starting Adriamycin treatment, one month and six months after treatment was 16.8 cm/s, 14 cm/s and 14 cm/s (p<0.001).” Confirm whether these values were normally distributed. If normally distributed, please include the 95% confidence intervals. If not, replace means with medians and interquartile ranges (IQR) for consistency and statistical appropriateness.

Please fix the formatting of Table 1 & 2 so they will be more readable (see screenshots). I believe it would be much easier to read Table 1 if it was formatted in the following way:

Baseline 1-Month 6-Months p-value

EF Median (IQR) Median (IQR) Median (IQR)

SM … … …

E’ … … …

E/A … … …

….

And Table 2 if it was formatted in the following manner:

Baseline 1-Month 6-Months p-value

E’*

≥ 8 cm/s Median (IQR) Median (IQR) Median (IQR)

< 8 cm/s … … …

… … …

Figures

• Figure 1: Create a figure comparing baseline measurements with measurements taken at either 1 or 6 months post-treatment. The current figure is too generic and less informative with respect to the study’s objectives. Consider including a visual comparison demonstrating echocardiographic doppler changes before and after treatment (as also suggested in the previous manuscript review).

• Figure 2: Adjust the y-axis to start at ‘y=0’. The current axis range artificially exaggerates the perceived effect.

Discussion

The discussion requires significant editing to eliminate repetition and redundancy. Revise this section to clearly and concisely include:

• One paragraph summarizing your key findings.

• One paragraph discussing relevant literature.

• One paragraph highlighting the strengths of this study compared to prior research.

• One paragraph clearly outlining limitations.

• A brief summary or conclusion statement.

The discussion section should be much shorter than its current form.

Ensure consistent citation formatting, using brackets ([]) rather than parentheses (()). For example, revise: “In alignment with these findings, our study observed early diastolic dysfunction in 80% of patients at one-month post-treatment (16),” to “In alignment with these findings, our study observed early diastolic dysfunction in 80% of patients at one-month post-treatment [16].”

**Do you want your identity to be public for this peer review?** For information about this choice, including consent withdrawal, please see our Privacy Policy

Reviewer #1: No

---

## [Author Response · Author response to Decision Letter 2]

24 May 2025

Review (RR1) – Early Detection of Cardiotoxicity in Children Receiving Adriamycin: A Comparison of Tissue Doppler and Standard Echocardiography with Troponin I as an Early Marker

I thank the authors for addressing the previous comments and making notable improvements to the manuscript. However, there remain some important areas needing further clarification and revision to enhance the manuscript's overall quality. Specific points are detailed below.

Introduction

No comments

Methods

“Diastolic Function: Normal diastolic function was defined as an E/A ratio of 0.9–1.5. Abnormal diastolic function was considered if the E/A ratio was <0.9 or >2.” Clarify how values between 1.5 and 2 were categorized, as the provided ranges do not currently cover these values. Did you intend the normal range to be 0.9–2?

Response. Thank you for your constructive comments. According to referee comment, normal diastolic function was defined as an E/A ratio of 0.9–2.

All echocardiographic variables you reference (LVEF, TAPSE, S’, E/E’, E/A) vary significantly by age across the 2–14 year range in your study. Applying adult-reference ranges universally is a substantial limitation and should be acknowledged explicitly in the discussion. You may, however, also mention that since comparisons were intra-patient (patients compared to themselves), then this limitation likely does not significantly impact your main results.

Response. In light of your insightful feedback, we presented more detailed information in end of discussion section.

Results

“The median follow-up duration for cardiac evaluation was six months.” Given that all patients had a minimum follow-up of 6 months, stating the median as exactly 6 months is unclear or possibly incorrect. Consider revising to: "All patients were followed for the 6-month duration of the study, and no further follow-up data were available," or clearly state the actual median and IQR for available follow-up durations.

Response. Thank you for your valuable comment. All our patients were followed for the 6-month duration of the study, and no further follow-up data were available. Therefore, we revised the sentence in Result section (end of Demographics section).

“Standard Doppler Echocardiography:” Please correct the formatting to properly align the section title. (remove the bullet-point format and align the paragraph with the rest of the text).

Response. According to referee comment, it was corrected.

The presentation of the statistical analysis results is currently difficult to follow. Clarify precisely what EF1, EF2, and EF3 represent. Consider revising to: "The Kolmogorov–Smirnov test indicated significant deviation from a normal distribution (statistic > 0.153, p < 0.05). The median EF at baseline, one month, and six months’ post-treatment were 65% (range: 50–78%), 65% (range: 50–75%), and 62% (range: 50–76%), respectively."

Response. So thankful for your comment. According to referee comment, the presentation of the statistical analysis results was revised and clarified.

“The mean S’ velocity before starting Adriamycin treatment, one month after treatment, and six months after treatment was 10.3 cm/s, 8.3 cm/s, and 8.3 cm/s,” and similarly, “The mean E’ velocity before starting Adriamycin treatment, one month and six months after treatment was 16.8 cm/s, 14 cm/s and 14 cm/s (p<0.001).” Confirm whether these values were normally distributed. If normally distributed, please include the 95% confidence intervals. If not, replace means with medians and interquartile ranges (IQR) for consistency and statistical appropriateness.

Response. Thank you for your constructive comments. In present study, a 95% confidence interval was used for the standard normal distribution. Whereas, data that do not follow a normal distribution was presented as medians and interquartile ranges (IQR). According to referee comment, the correction was made in the text and table.

Please fix the formatting of Table 1 & 2 so they will be more readable (see screenshots). I believe it would be much easier to read Table 1 if it was formatted in the following way:

Response. In light of your insightful feedback, we fixed the formatting of Table 1 & 2 and they are more readable. Table 1 and Table 2 were formatted in the following way.

Baseline 1-Month 6-Months p-value

EF Median (IQR) Median (IQR) Median (IQR)

SM … … …

E’ … … …

E/A … … …

….

And Table 2 if it was formatted in the following manner:

Baseline 1-Month 6-Months p-value

E’*

≥ 8 cm/s Median (IQR) Median (IQR) Median (IQR)

< 8 cm/s … … …

… … …

Figures

Figure 1: Create a figure comparing baseline measurements with measurements taken at either 1 or 6 months post-treatment. The current figure is too generic and less informative with respect to the study’s objectives. Consider including a visual comparison demonstrating echocardiographic doppler changes before and after treatment (as also suggested in the previous manuscript review).

Figure 2: Adjust the y-axis to start at ‘y=0’. The current axis range artificially exaggerates the perceived effect.

Response. So thankful for your comment. Since according to the reviewer, Tables were made more readable and complete, therefore all the information in the tables was presented; we removed the figures from the manuscript.

Discussion

The discussion requires significant editing to eliminate repetition and redundancy. Revise this section to clearly and concisely include:

• One paragraph summarizing your key findings.

• One paragraph discussing relevant literature.

• One paragraph highlighting the strengths of this study compared to prior research.

• One paragraph clearly outlining limitations.

• A brief summary or conclusion statement.

The discussion section should be much shorter than its current form.

Response. So thankful for your comment. Based on the reviewer's comments, the discussion was rewritten based on the comments provided (more concise key findings, addressing relevant literature, highlighting strengths of the study and comparisons with previous research, clearly stating limitations, and stating comprehensive conclusions). Finally, repetitive sections were removed and the discussion section was presented in a shorter and clearer manner.

Ensure consistent citation formatting, using brackets ([]) rather than parentheses (()). For example, revise: “In alignment with these findings, our study observed early diastolic dysfunction in 80% of patients at one-month post-treatment (16),” to “In alignment with these findings, our study observed early diastolic dysfunction in 80% of patients at one-month post-treatment [16].”

Response. Thank you for your valuable feedback. We were ensured consistent citation formatting, using brackets ([]).

---

## [Decision Letter · Decision Letter 2]

18 Jun 2025

Dear Dr. Oveisi,

Thank you for submitting your manuscript to PLOS ONE. After careful consideration, we feel that it has merit but does not fully meet PLOS ONE’s publication criteria as it currently stands. Therefore, we invite you to submit a revised version of the manuscript that addresses the points raised during the review process.

Very good progress has been made in last two rounds of review, before I accept your manuscript for publication please make the following minor adjustments:

**1)** Please restate this paragraph in more informal way since these numbers can be found in Table 1:

"After testing the normality distribution of data,

Kolmogorov–Smirnov statistic more than 0.153 with a p-value <0.05 indicated significant

deviation from normality at α = 0.05: EF1 of baseline; median= 65 (50, 78), IQR= 10, EF2 of

after 1 month; median= 65 (50, 75), IQR= 8.25, and EF of after 6 months; median= 62 (50, 76),

IQR= 7.5. Therefore, there was significant changes in LVEF one and six months post-treatment

by Friedman test p<0.001."

**2)** When giving the IQR of certain variables (e.g. IQR) please give it in format with 25th and 75th percentile, not as single number since IQR can be asymmetric and that is not captured well by a single number.

I am looking forward to revised version of manuscript.

We look forward to receiving your revised manuscript.

Kind regards,

Benjamin Benzon, Ph.D., M.D.

Academic Editor

PLOS ONE

Journal Requirements:

**Additional Editor Comments :**

Very good progress has been made in last two rounds of review, before I accept your manuscript for publication please make the following minor adjustments:

1) Please restate this paragraph in more informal way since these numbers can be found in Table 1:

"After testing the normality distribution of data,

Kolmogorov–Smirnov statistic more than 0.153 with a p-value <0.05 indicated significant

deviation from normality at α = 0.05: EF1 of baseline; median= 65 (50, 78), IQR= 10, EF2 of

after 1 month; median= 65 (50, 75), IQR= 8.25, and EF of after 6 months; median= 62 (50, 76),

IQR= 7.5. Therefore, there was significant changes in LVEF one and six months post-treatment

by Friedman test p<0.001."

2) When giving the IQR of certain variables (e.g. IQR) please give it in format with 25th and 75th percentile, not as single number since IQR can be asymmetric and that is not captured well by a single number.

I am looking forward to revised version of manuscript.

Reviewers' comments:

Reviewer's Responses to Questions

**Comments to the Author**

Reviewer #1: All comments have been addressed

2. Is the manuscript technically sound, and do the data support the conclusions?

Reviewer #1: Yes

3. Has the statistical analysis been performed appropriately and rigorously?

Reviewer #1: Yes

4. Have the authors made all data underlying the findings in their manuscript fully available?

Reviewer #1: No

5. Is the manuscript presented in an intelligible fashion and written in standard English?

Reviewer #1: Yes

Reviewer #1: Thank you for addressing all my comments. Two small things that should be changed before publications:

1. I would remove the text you added "A 95% confidence interval was used for the standard normal distribution. Whereas, data that do not follow a normal distribution was presented as medians and interquartile ranges (IQR) (Table 1)." -- this should be in methods and not results. In the results I only meant that when you mention numbers in the text, you should include in parenthesis the 95% CI. You don't need to describe the method used here, just include the resulting numbers.

2. Work with the editorial office on the formatting of the tables. For example, change 'befor' in table 1 to 'before', and either remove all the asterisks there or include the description of each an each sequence of asterisks (e.g. what does *, **, *** etc. mean..).

Thank you

**Do you want your identity to be public for this peer review?** For information about this choice, including consent withdrawal, please see our Privacy Policy

Reviewer #1: No

---

## [Author Response · Author response to Decision Letter 3]

30 Jun 2025

Dear editors and reviewers,

Thank you very much for sending us the reviewers’ comments on our manuscript entitled “Early Detection of Cardiotoxicity in Children Receiving Adriamycin: A Comparison of Tissue Doppler and Standard Echocardiography with Troponin I as an Early Marker”. We appreciate the time and effort that the editors and the reviewers have dedicated to providing the valuable feedback on our manuscript. We are grateful to the reviewers for their insightful comments. We have incorporated changes to reflect the suggestions provided by the reviewers and improved the logic and the language of the manuscript. The revised contents are highlighted in blue within the revised manuscript.

Our point-by-point responses to the reviewers’ comments are as follows:

Additional Editor Comments :

Very good progress has been made in last two rounds of review, before I accept your manuscript for publication please make the following minor adjustments:

1) Please restate this paragraph in more informal way since these numbers can be found in Table 1:

"After testing the normality distribution of data,

Kolmogorov–Smirnov statistic more than 0.153 with a p-value <0.05 indicated significant

deviation from normality at α = 0.05: EF1 of baseline; median= 65 (50, 78), IQR= 10, EF2 of

after 1 month; median= 65 (50, 75), IQR= 8.25, and EF of after 6 months; median= 62 (50, 76),

IQR= 7.5. Therefore, there was significant changes in LVEF one and six months post-treatment

by Friedman test p<0.001."

2) When giving the IQR of certain variables (e.g. IQR) please give it in format with 25th and 75th percentile, not as single number since IQR can be asymmetric and that is not captured well by a single number.

Authors’ answer: Thank you very much for your suggestion. We rewrite this sentence and table1.

“Due to the data not adhering to a normal distribution (Kolmogorov–Smirnov test, p < 0.05), LVEF values were presented as medians and interquartile ranges. Table 1 illustrates LVEF slightly declined over time, and the Friedman test confirmed a significant change at one and six months after Adriamycin treatment (p < 0.001).”

Reviewer #1: Thank you for addressing all my comments. Two small things that should be changed before publications:

1. I would remove the text you added "A 95% confidence interval was used for the standard normal distribution. Whereas, data that do not follow a normal distribution was presented as medians and interquartile ranges (IQR) (Table 1)." -- this should be in methods and not results. In the results I only meant that when you mention numbers in the text, you should include in parenthesis the 95% CI. You don't need to describe the method used here, just include the resulting numbers.

Authors’ answer: Thank you very much for drawing attention to this. We fixed it.

2. Work with the editorial office on the formatting of the tables. For example, change 'befor' in table 1 to 'before', and either remove all the asterisks there or include the description of each an each sequence of asterisks (e.g. what does *, **, *** etc. mean..).

Authors’ answer: Thank you very much for your suggestion. We fixed it.

---

## [Editor Report · Decision Letter 3]

4 Jul 2025

Early Detection of Cardiotoxicity in Children Receiving Adriamycin: A Comparison of Tissue Doppler and Standard Echocardiography with Troponin I as an Early Marker

PONE-D-25-05750R3

Dear Dr. Oveisi,

We’re pleased to inform you that your manuscript has been judged scientifically suitable for publication and will be formally accepted for publication once it meets all outstanding technical requirements.

Kind regards,

Benjamin Benzon, Ph.D., M.D.

Academic Editor

PLOS ONE

Additional Editor Comments (optional):

Congratulations, thank you for your effort and cooperation in extensive revisions that were made during the review process.
---

## [Editor Report · Acceptance letter]

PONE-D-25-05750R3

PLOS ONE

Dear Dr. Oveisi,

I'm pleased to inform you that your manuscript has been deemed suitable for publication in PLOS ONE. Congratulations! Your manuscript is now being handed over to our production team.

Kind regards,

on behalf of

Dr. Benjamin Benzon

Academic Editor

PLOS ONE